# A Central Role of Telomere Dysfunction in the Formation of a Unique Translocation within the Sub-Telomere Region Resulting in Duplication and Partial Trisomy

**DOI:** 10.3390/genes13101762

**Published:** 2022-09-29

**Authors:** Radhia M’Kacher, Marguerite Miguet, Pierre-Yves Maillard, Bruno Colicchio, Sophie Scheidecker, Wala Najar, Micheline Arnoux, Noufissa Oudrhiri, Claire Borie, Margaux Biehler, Andreas Plesch, Leonhard Heidingsfelder, Annelise Bennaceur-Griscelli, Alain Dieterlen, Philippe Voisin, Steffen Junker, Patrice Carde, Eric Jeandidier

**Affiliations:** 1Cell Environment DNA Damage R&D, Genopole, 91058 Evry, France; 2Laboratoire de Génétique, Groupe Hospitalier de la Région de Mulhouse Sud-Alsace, 68070 Mulhouse, France; 3Service de Génétique Hôpitaux Universitaires de Strasbourg, Hôpital de Haute Pierre, 1, Rue Molière, 67000 Strasbourg, France; 4IRIMAS, Institut de Recherche en Informatique, Mathématiques, Automatique et Signal, Université de Haute-Alsace, 68070 Mulhouse, France; 5Laboratoire de Diagnostic Génétique, Hôpitaux Universitaires de Strasbourg, Nouvel Hôpital Civil, 1, Place de l’Hôpital, 67000 Strasbourg, France; 6APHP-Service d’Hématologie-Oncohématologie Moléculaire et Cytogénétique Hôpital Paul Brousse Université Paris Saclay, 94801 Villejuif, France; 7MetaSystems GmbH, Robert-Bosch-Str. 6, 68804 Altlussheim, Germany; 8Institute of Biomedicine, University of Aarhus, 8000 Aarhus, Denmark; 9Department of Hematology Gustave Roussy Cancer Campus, Paris Saclay, 94805 Villejuif, France

**Keywords:** postnatal, telomere, sub-telomere region, duplication, trisomy, chromosomal instability

## Abstract

Telomeres play a major role in maintaining genome stability and integrity. Putative involvement of telomere dysfunction in the formation of various types of chromosomal aberrations is an area of active research. Here, we report a case of a six-month-old boy with a chromosomal gain encompassing the 11q22.3q25 region identified by SNP array analysis. The size of the duplication is 26.7 Mb and contains 170 genes (OMIM). The duplication results in partial trisomy of the region in question with clinical consequences, including bilateral renal dysplasia, delayed development, and a heart defect. Moreover, the karyotype determined by R-banding and chromosome painting as well as by hybridization with specific sub-telomere probes revealed the presence of an unbalanced t(9;11)(p24;q22.3) translocation with a unique breakpoint involving the sub-telomere region of the short arm of chromosome 9. The karyotypes of the parents were normal. Telomere integrity in circulating lymphocytes from the child and from his parents was assessed using an automated high-throughput method based on fluorescence in situ hybridization (FISH) with telomere- and centromere-specific PNA probes followed by M-FISH multicolor karyotyping. Very short telomeres, as well as an increased frequency of telomere loss and formation of telomere doublets, were detected in the child’s cells. Interestingly, similar telomere profiles were found in the circulating lymphocytes of the father. Moreover, an assessment of clonal telomere aberrations identified chromosomes 9 and 11 with particularly high frequencies of such aberrations. These findings strongly suggest that telomere dysfunction plays a central role in the formation of this specific unbalanced chromosome rearrangement via chromosome end-to-end fusion and breakage–fusion–bridge cycles.

## 1. Introduction

Children with unbalanced genomic aberrations present with variable phenotypes that include three relatively common manifestations: developmental delay, intellectual disability, and malformation of the heart [1,2,3,4]. Thus, precise and sensitive evaluation of these aberrations is an important challenge to clinicians offering genetic counseling to families, in particular prenatal counseling. Although substantial progress in the identification of genomic aberrations and their clinical manifestations has been made during the last few decades [5], the detection of repeated sequences such as telomere and centromere sequences is underestimated using standard cytogenetic techniques [6]. Consequently, crucial information is lacking not only on the analysis of mechanisms underlying the formation of these aberrations but also on the handling of the resulting mental and physical complications of these patients on their follow-up.

Telomere instability is a biological factor that contributes to the formation of genomic aberrations [7]. Telomere DNA consists of long hexamer iterations (TTAGGG)n, which are responsible for the integrity and stability of the genome [8]. In normal somatic cells, telomere DNA is shortened by 50–220 bp during each cell division [9,10]. When telomeres become too short, cells stop dividing and ultimately enter senescence [11]. Hence, telomeres are considered a cellular clock controlling cell division and cell death [12]. Telomere dysfunction is considered one of the major mechanisms causing chromosomal instability occurring through breakage/fusion/bridge (B/F/B) cycles, described first by McClintock [13]. Thus, when telomeres are lost from a replicated chromosome, the ends of its sister chromatids may fuse. If the centromeres during anaphase are pulled into opposite directions, the fused sister chromatids will break unequally. This process will be repeated in the subsequent cell cycles until the chromosome acquires new telomeres securing its stability, however, at the cost of a heavy genetic burden [14]. Thus, B/F/B cycles have been described frequently during the course of tumor progression [7]. In clinical genetic settings, analysis of telomere length and stability has not yet been introduced as a routine technique in the elucidation of prenatal and postnatal karyotypes. Therefore, the process of B/F/B cycles has not been addressed in the mechanisms of chromosomal aberrations in this group of patients.

Recently, we demonstrated that the introduction of telomere and centromere (TC) staining followed by M-FISH (TC+M-FISH) not only renders analysis of chromosomal aberrations more efficient and robust but also permits quantification of telomere lengths and aberrations [6].

Here, we have applied this technique to identify the possible origin of a 26.7 Mb duplication of 11q22.3q25 in a six-month-old boy. We propose a causal role of the telomere instability identified in the parents that contributes to the formation of this duplication via breakage fusion bridge cycles. We discuss the importance of telomere analysis in prenatal as well as in postnatal diagnosis.

## 2. Materials and Methods

### 2.1. Materials

Blood samples were obtained from a six-month-old boy and from his parents (father 36 years old; mother 36 years old). A cohort of 150 healthy donors was included as controls. Cytogenetic slides were prepared according to previously published protocols [15].

Informed consent, according to the Declaration of Helsinki, was obtained from patients and their parents. Analyses were performed in the context of current diagnosis explorations.

Informed consent was collected from healthy donors (ethical approval code: Comités de protection des personnes CPP 97/06 and Nephrovir study).

### 2.2. Methods

#### 2.2.1. Molecular Genetics

DNA from blood samples was extracted using the DNEasy Blood Kit (QIAGEN, Courtabouf, France).

Microarray analysis was carried out using an SNP array with approximately 300,000 SNP across the genome (HumanCyto SNP-12 Illumina^®^, Illumina, Evry, France) with a practical average resolution of 70 kb. Data were analyzed with GenomeStudio v.2011.1 software (genome build hg19).

#### 2.2.2. Preparation of Metaphase Spreads

Cultured lymphocytes were exposed to colcemid (0.1 µg/mL) (Gibco KaryoMAX, ThermoFisher, Waltham, MA, USA) for 1 h at 37 °C, 5% CO_2_, in a humidified atmosphere to block them in metaphase. After cell harvest and centrifugation for 7 min at 1400 rpm at room temperature (RT), the supernatant was removed and the cells were re-suspended in a warm (37 °C) solution of 0.075 M KCl (Merck, Kenilworth, NJ, USA) and incubated for 20 min in a 37 °C water bath (hypotonic shock). For pre-fixation of the cells, approximately five drops of fixative (3:1 ethanol/acetic acid) were added to each tube under gentle permanent agitation and the tubes were centrifuged for 7 min at 1400 rpm at RT. The supernatant was removed and the cells were suspended in fixative solution and centrifuged as before. After two additional rounds of fixation, cells were stored in fixative solution at 4 °C overnight and metaphases were spread on cold, wet slides the next day. Slides with metaphase spreads were dried overnight at RT and stored at −20 °C until further use.

#### 2.2.3. Cytogenetics Investigation

R-banding of chromosomes for karyotyping and fluorescence in situ hybridization was carried out using standard procedures. Specific probes for sub-telomere regions of short and long arms of chromosome 9 (tel9p: RH65569, tel9q: D9S2168) and 11 (tel11p: D11S2071, tel11q: D11S4974) (Cytocell, Sysmex, Baltimore, MD, USA) and for whole chromosome 11 painting (XCP11) (MetaSystems, Altlussheim, Germany) were employed.

#### 2.2.4. Quantification of Telomere Length and Assessment of Telomere Aberrations

Telomere and centromere staining followed by M-FISH was performed on cytogenetic slides as described previously in M’kacher et al. [6]. Telomeres were hybridized with a Cy-3-labelled PNA probe specific for (TTAGGG) and centromeres with a FITC-labelled probe specific for centromere sequences. The M-FISH technique was applied after the capture of telomere and centromere signals.

Quantification of telomeres in nuclei was performed using a high-throughput automated process (TeloScore, Cell Environment, Evry, France) that makes it possible to quantify mean telomere FISH signals and their heterogeneity as well as their intercellular variation in a vast number of cells [6]. The mean fluorescence intensity signals (FI) of telomeres strongly correlated with the telomere lengths estimated by conventional Southern blot analysis using the telomeric restriction fragment (TRF) (R2 = 0.721 and *p* = 2.128 × 10^−8^). The mean telomere length is expressed in kb.

Telomere aberrations were classified as (1) telomere loss, (2) telomere doublets, and (3) telomere deletion. An average of 50 metaphases of each individual was scored for these analyses.

## 3. Results

### 3.1. Case Presentation

A six-month-old boy presented with developmental delay, bilateral renal dysplasia, and malformation of the heart. Genomic analysis of the proband using an SNP array revealed the presence of a partial trisomy 11q22.3q25 (Figure 1) encompassing a 26.7 Mb duplication with 170 genes (OMIM).

Conventional cytogenetics using R-banding demonstrated the presence of an unbalanced translocation (Figure 2A). Chromosome painting confirmed this translocation: der(9)t(9;11)(p24;q22) (Figure 2B). The breakpoint in chromosome 9 was localized to the repeated sequence of the telomere region. The presence of sub-telomeres of 9p was confirmed (Figure 2C). It should be added that the SNP analysis of chromosome 9 did not reveal any alterations (Figure 1). Furthermore, the karyotypes of both parents were normal.

### 3.2. Telomere Analysis

To understand the origin and the mechanism underlying the formation of this partial trisomy, telomere length and telomere aberrations were assessed in the proband and in his parents using telomere and centromere-specific probes followed by M-FISH (Figure 3).

This approach allows the quantification of telomere signals in the nuclei as well as in metaphases. The mean telomere length of the father was 6.91 kb, which is a significant reduction compared to controls at a similar age (<45 years and >35 years) (*p* = 1.8310^−6^) (Figure 4A). The mean telomere length of the proband was 8.5 kb, which is also a significant reduction compared to that of healthy donors (<5 years) (*p* = 5 × 10^−4^) (Figure 4A). The mean telomere length of the mother was 8.69 kb. Our approach also permits detailed quantification of telomere length in individual cells, thus making it possible to assess the frequency of cells with extreme telomere shortening (<5 kb). The frequency of such cells was significantly higher in the father (16.4%) and in the mother (8.6%) than that observed in healthy donors (3.2%) with similar telomere length (*p* < 10^−16^) (Figure 4B).

Next, we determined the telomere phenotypes in cells of the proband, of his parents, and of donors of similar age. Telomere aberrations are classified into three groups: (i) Telomere loss is defined as a signal-free end at a single chromatid, an aberration that leads to telomere end fusion and breakage/fusion/bridge cycles [6]; (ii) telomere doublets or telomere fragility are defined as more than one telomere signal at a single arm, an aberration signaling inadequate telomere replication and dysfunction of the shelterin proteins [6]; (iii) telomere deletion is defined as the loss of two telomere signals on the same arms, an aberration considered to represent double-strand breaks, leading to activation of DNA repair mechanisms.

In metaphases of the proband and his parents, telomere loss was significantly more extensive than that in healthy donors (0.5 per cell for the proband, 1.7 per cell for the father, 0.5 per cell for his mother, and 0.2 per cell for the healthy donors, *p* < 10^−15^) (Figure 5B). Similar results were obtained on the frequencies of formation of telomere doublets that are considered a marker of fragility of telomeres (13.6/cell for the proband, 15.6/cell for the mother, 11.8/cell for the father, and 4.5/cell for healthy donors, *p* < 10^−3^) (Figure 5A,B). The frequencies of telomere deletion are much lower and similar to that observed in controls. 

### 3.3. Correlation between Telomere Instability and the Formation of Chromosomal Rearrangements

In order to assess a putative correlation between telomere instability and the occurrence of chromosomal aberrations, we calculated the frequencies of telomere loss and of telomere doublets within each chromosome from both the proband and his parents. Significantly higher frequencies of telomere losses and of telomere doublets were identified in chromosomes 11 and 9 than in any other chromosomes (*p* < 10^−3^) of the parent (Figure 6). This finding is striking because these two particular chromosomes take part in the duplication and translocation events in the proband.

It should be mentioned that also micronucleus formation and non-clonal aberrations, such as centric ring and dicentric chromosomes, were observed in circulating lymphocytes of the father (Figure 7) but not in the mother or in the proband. These findings strongly suggest a central role of telomere dysfunction in the formation of micronuclei and of non-clonal chromosome aberrations. Moreover, our findings emphasize the clinical relevance of scoring also micronuclei and non-clonal aberrations in analyses of chromosomal aberrations.

## 4. Discussion

The consequences of telomere shortening and dysfunction in normal and pathological conditions have been extensively studied. Thus, telomeres play complex roles not only in cellular processes such as cellular senescence and apoptosis but also in transcriptional regulation [16], chromosome stability [17], mitochondrial function [18], meiotic regulation [19], and cellular differentiation [20]. Recently, we have demonstrated that telomere shortening and high frequencies of telomere aberrations are common characteristics of certain genetic disorders, including infertility disorders [21].

However, analysis of telomere instability has not yet been implemented in the clinical genetics laboratory as a routine analysis in either prenatal or postnatal diagnostics. This may be ascribed to the fact that conventional cytogenetic techniques, as well as molecular genetic approaches, are not suitable for the detection of repeated sequences such as telomeres. Furthermore, the sensitivity of some of the techniques used for quantification of telomere length is inadequate. These technical challenges may, in part, explain some of the conflicting data published on measurements of telomere lengths in newborns with genetic disorders such as Down syndrome [22,23] and Turner syndrome [24,25].

Thus, there is an urgent need for standardization of the techniques employed for the quantification of telomeres and a need for establishing a publically available database for telomeres.

To meet these fundamental requirements, we recently developed an automated approach for the quantification of telomere length in vast numbers of single cells. The technique is based on cytogenetic preparations and fluorescence in situ hybridization (FISH) with specific probes for telomeres and centromeres. The technique is robust, reliable, and reproducible [6,15]. With this approach, we have previously shown that telomeres in cells of patients with certain genetic syndromes were significantly shorter than those in cells of healthy controls of similar age [26]. Furthermore, the frequencies of patient cells with drastic telomere shortening were much higher compared with cells of healthy controls [6]. In the patient cells, we also found higher rates of telomere aberrations resulting in augmented telomere shortening, possibly leading to the progression of chromosome instability [27,28].

In the present study, we have applied our approach as a proof of principle in a clinical setting to assess telomere dysfunction in a six-month-old boy presenting with an unbalanced translocation. We wished to validate not only the additional benefits of investigating telomere dysfunction in postnatal diagnostic workups in clinical genetics but also the feasibility of combining such an analysis with the identification of chromosomal aberrations.

Conventional R-banding and chromosome painting of metaphases of the proband’s cells identified an unbalanced translocation der(9)t(9;11)(p24;q22.3) with a unique breakpoint involving the repeated telomere sequences region of the short arm of chromosome 9. The subtelomere region had been preserved in chromosome 9 with an absence of only telomere repeated sequences in the breakpoint, thus suggesting that this aberration is related to telomere deletion. Moreover, genomic analysis using SNP arrays revealed the presence of a partial trisomy 11q22.3q25 encompassing a 26.7 Mb duplication.

The karyotypes of both parents were normal. However, quantitative analysis of cells of the father revealed telomere shortening and high rates of telomere loss. Although the telomere lengths of the mother were comparable to those of healthy donors of similar age, a high frequency of telomere doublets was found in her circulating lymphocytes and also in those of the father, although at lower frequencies.

Remarkably, a detailed analysis of telomere aberrations in individual chromosomes of the proband and his parents showed that the higher rate of both telomere loss and deletion in cells of the father and of the mother occurred on chromosomes 9 and 11, i.e., those two chromosomes implicated in the translocation in the proband. Therefore, an association between a particular telomere fragility observed on chromosomes 9 and 11 in the mother and telomere loss in the same chromosomes in the father may constitute an increased risk of a translocation event. In addition, the high rate of micronucleus formation and non-clonal aberrations in the circulating lymphocytes of the father emphasizes the central role of telomere dysfunction in the progression of chromosomal instability [29].

We hypothesize that loss of telomere functionality in chromosomes 9 and 11 allows chromosome end-to-end fusion and subsequent activation of breakage/fusion/bridge (B/F/B) cycles that will be repeated for several cell divisions. Ultimately, these events result in DNA duplications and unbalanced chromosomal rearrangements. A schematic representation of our hypothesis on the formation of the duplication and the unbalanced translocation in the proband after the loss of telomeres is outlined in Figure 8.

Our findings are in accordance with previously published data and support the notion that high rates of formation of micronuclei [30,31] as well as of non-clonal aberrations such as dicentric chromosomes are superior biomarkers of chromosomal instability in patients with genetic disorders and in their parents with no clinical symptoms. Our study extends our insights into the origin of this kind of genomic instability.

To our knowledge, this is the first study reporting a putative causal association between telomere instability in parents and the formation of a specific de novo chromosome aberration in their offspring.

However, these novel findings also call for a need for further analyses of the role of telomere dysfunction in our understanding of the origin of prenatal (post-conception) chromosome aberrations. The introduction of these analyses in the clinical setting is warranted for the follow-up of chromosomal anomalies found in parents and their newborns.

## 5. Conclusions

The main result of our study is the identification of telomere instability in parents with normal karyotypes and the implication of this telomere instability in the formation of a chromosomal rearrangement in their offspring.

Our findings strongly suggest that telomere instability in parents may play a causal role in the formation of chromosomal aberrations and in the progression of chromosomal instability of their offspring. Further analysis will be needed to validate this hypothesis. Genomic analysis of this region of chromosome 11 (11q22.3) could shed light on the possible mechanisms.

Collectively, this data and our previous findings [6,21,32,33] underscore the need for further investigations on telomeres and their putative association to chromosomal aberrations in the clinical genetics setting. We also propose that telomere analysis (length and aberrations) be included in future prenatal diagnosis as a routine procedure in the establishment of karyotypes. For these investigations, the agreement on a standard cut-off of telomere aberrations and the generation of a database of telomere lengths and aberrations are mandatory.

## Figures and Tables

**Figure 1 genes-13-01762-f001:**
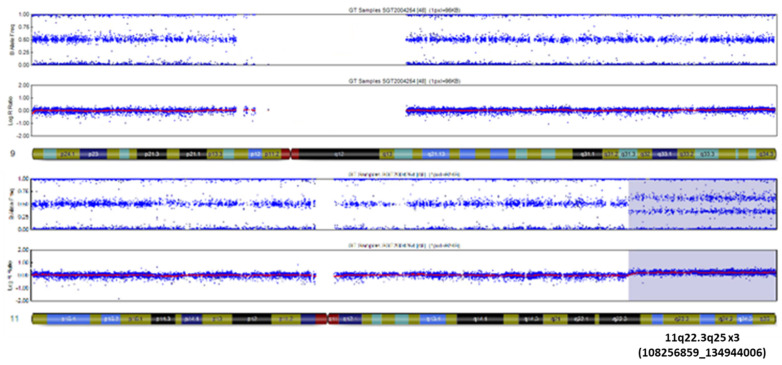
Genomic analysis of proband using SNP array reveals a 26.7 Mb duplication of 11q22.3q25.

**Figure 2 genes-13-01762-f002:**
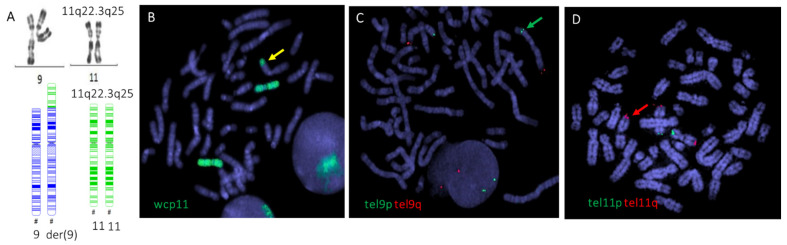
Identification of der(9)t(9;11)(p24;q22)(D11S4974+,wcp11+,RH65569+,D9S2168+) in metaphases of the proband using conventional and molecular cytogenetics. (**A**) Conventional R-banding reveals the presence of the unbalanced translocation. (**B**) Chromosome 11 painting (green) confirms the translocation (yellow arrow). (**C**) FISH with chromosome-specific sub-telomere probes demonstrates that the sub-telomere region, containing the tel9q (RH65569) marker (green arrow), within the derivative chromosome 9 short arm, is preserved. (**D**) Sub-telomere sequences of 11q (D11S4974) are present at the end of the short arm of the derivative chromosome 9 (red arrow). Sub-telomeres for the q region are in red, and the sub-telomere p region is in green. Metaphases were counterstained with DAPI. One hundred metaphases were analyzed, and twenty metaphases were classified.

**Figure 3 genes-13-01762-f003:**
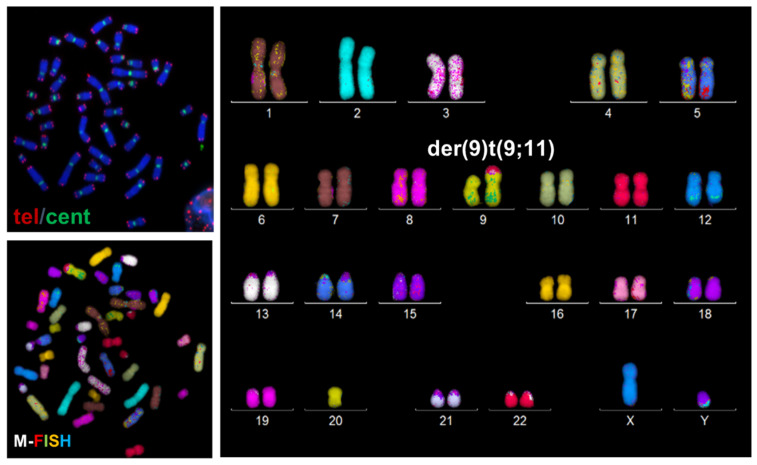
Telomere (red) and centromere (green) staining followed by M-FISH confirms the presence of this unbalanced translocation: der(9)t(9;11)(p24;q22.3).

**Figure 4 genes-13-01762-f004:**
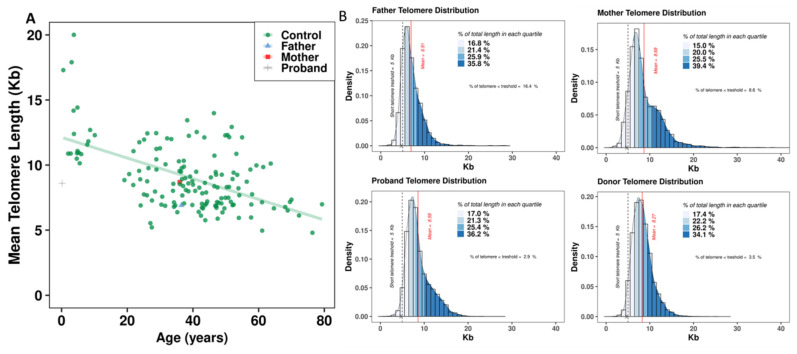
Quantification of telomere length in interphase lymphocytes of the proband and his parents. (**A**) Telomere length (kb) as a function of age in a large cohort of lymphocytes from healthy donors (150 donors, mean age 36 years, range 0.5–79 years) (red circles), in the father (blue triangle), in the mother (red square), and in the proband (purple cross). A linear regression of biological telomere loss with the age of healthy donors is presented. A significant difference between the telomere length of the proband and of controls with similar age (<5 years) is noted. The telomere length of the father is significantly reduced compared to that in healthy donors with similar age (<45 years and >35 years) within the lower limit. (**B**) Global quantification of telomere signals in nuclei of 10,000 cells of the proband, his parents, and of a healthy donor. The method allows estimation of mean telomere length and the frequency of cells with extreme telomere shortening (<5 kb). A higher rate of these cells was observed in the father (16.4% in the father compared to 8.6% in the mother, 2.9% in the proband, and 3.5% in controls of similar age).

**Figure 5 genes-13-01762-f005:**
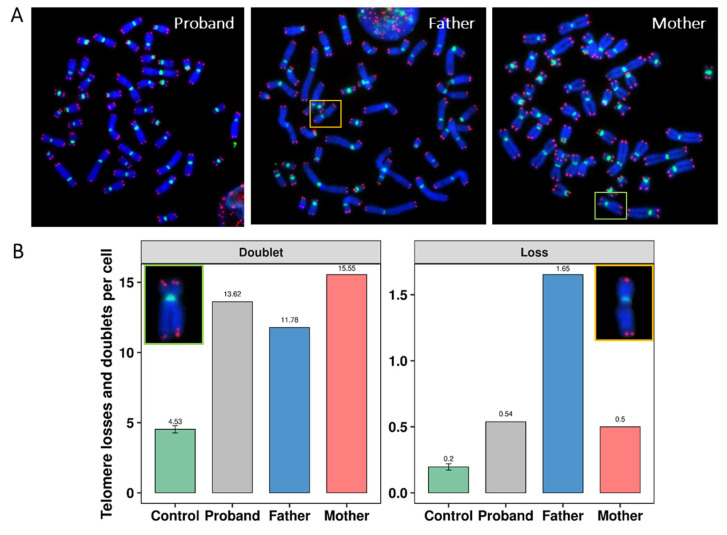
Detection of telomere aberrations by assessing telomeres on each chromosome in lymphocyte metaphases. (**A**) FISH on metaphase from the proband, the father, and mother using probes specific for telomere (red) and centromere (green) sequences, respectively, showing multiple formations of telomere doublets (yellow insert) and loss (green insert) in chromosome 9 and 11, respectively. (**B**) The frequency of telomere losses and of telomere doublet formations per cell in the proband, his father, his mother, and in the control. A significant difference was observed between the frequencies in the parents and in the proband compared to those in the control (*p* < 10^−15^ and *p* < 10^−3^, respectively). One hundred metaphases were analyzed per case.

**Figure 6 genes-13-01762-f006:**
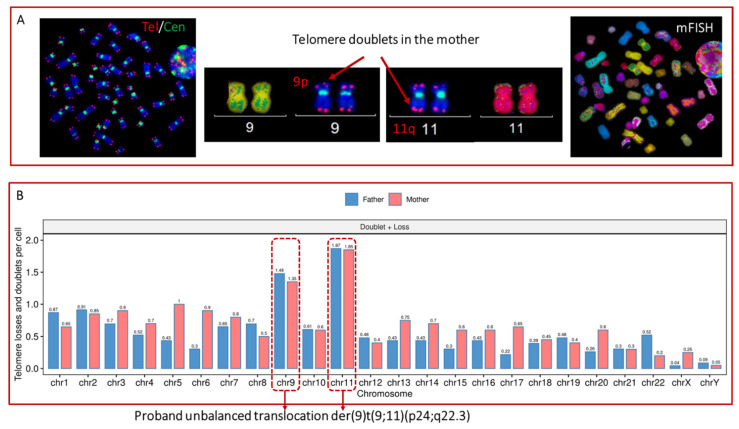
Analysis of telomere aberrations of individual chromosomes using telomere and centromere staining followed by M-FISH (**A**) metaphase from mother showing a high rate of telomere doublet formation. The short arm of chromosome 9 (9p) and the long arm of chromosome 11q exhibited a higher rate of telomere doublet, reflecting inadequate telomere replication and dysfunction of the shelterin proteins. (**B**) High rates of telomere loss and doublet formation in chromosomes 9 and 11 in the parents. Remarkably, these two chromosomes are implicated in the chromosomal aberration of the proband. Twenty metaphases were analyzed.

**Figure 7 genes-13-01762-f007:**
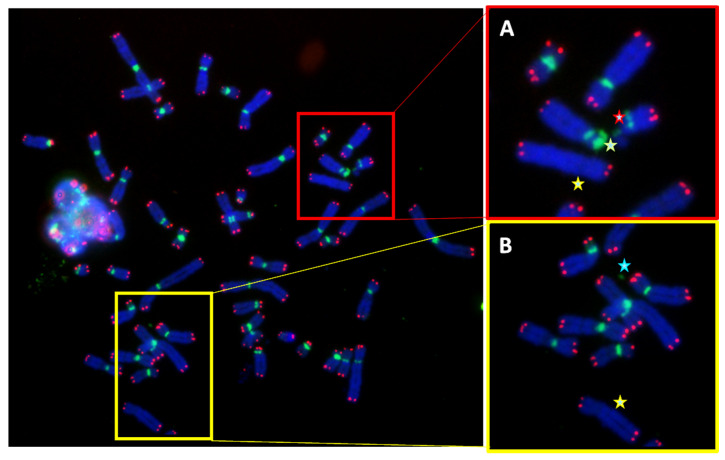
Metaphase of paternal circulating lymphocyte showing multiple non-clonal chromosomal aberrations. (**A**) Dicentric chromosome (green star), acentric chromosome (yellow star), and chromosome deletion (red star); (**B**) centric ring (blue star) and acentric chromosome (yellow star).

**Figure 8 genes-13-01762-f008:**
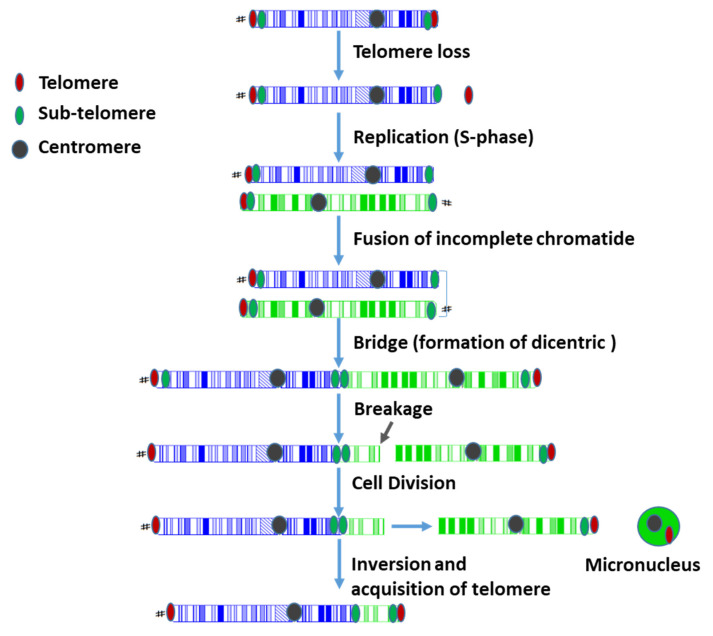
Schematic representation of a possible mechanism implicated in the formation of the chromosomal aberration observed in the proband. The breakage/fusion/bridge cycles are initiated after telomere loss. Chromosome 9 is depicted in blue, and chromosome 11 in green. See text for details.

## Data Availability

Not applicable.

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
