# Peer review of "A Central Role of Telomere Dysfunction in the Formation of a Unique Translocation within the Sub-Telomere Region Resulting in Duplication and Partial Trisomy"

_genes, 2022, doi:10.3390/genes13101762_

Round 1
Reviewer 1 Report
This is a really interesting paper. Elegantly written, well illustrated, and conceptually novel. The Authors present for the first time molecular cytogenetic evidence in support of a potentially causative association between telomeres instability in parents and the formation of de novo chromosome aberrations in their offspring. The findings are critically discussed to underscore the potential relevance of investigations of telomeres (and genomic instability) in the clinical genetic setting. I only have suggestions for a few, really minor amendments for the Authors to address, such as 1. spacing error in line 60; 2. a spelling mistake in line 71 as 'duting' should be 'during'; 3. in Figure 2 the fourth panel presents with no letter (should be D). The figure legend should be amended accordingly. Furthermore, what the yellow arrows are pointing at should be clarified for ease of reference; 4. In the legend of Figure 6, it should read 'yellow start', not 'star yellow', and so on for the other colours; 5. spacing error in line 289.
Author Response
Thank you very much for these comments
The minor corrections have been made
Reviewer 2 Report
Review Manuscript ID: 1905974
Title: A central role of telomere dysfunction in the formation of a unique translocation within the sub-telomere region resulting in duplication and partial trisomy.
It’s an interesting case where the telomeres in white blood cells from a six-month old boy harboring an unbalanced de novo der(9)t(9;11)(p24;q22.3), and his parents with normal karyotypes, were characterized. The authors suggest that this particular chromosomal rearrangement strongly is associated with telomere dysfunction.
There are some points that need clarification.
With respect to telomere analysis in section 3.2: What is the p-values for the proband and his parents to determine significance? In the figure legend to Fig. 3A it is stated that the telomere length of the father is within lower limits while in the body text it is stated to be significantly shorter. Please clarify.
With respect to telomere phenotypes (Fig. 4): Telomere aberrations are divided into three types but in Fig. 4 only two (doublets and loss) of these are depicted – what about the deletion type?
It would be helpful to show representative images of the different subtypes from each of the analyzed individuals. At the cellular and chromosomal level what is the impact of the different aberrant telomere subtypes and what do they reflect?
With respect to telomere analysis of individual chromosomes (Fig. 5): The x-axis indicates only normal chromosomes, which seems confusing because the proband has an unbalanced der(9)t(9;11), i.e. one normal chromosome 9 and one derivative. Further, it is not clear how this was handled when calculating frequency of telomere loss and doublet formation - the telomere on the der(9)t(9;11) p-arm is not of 9p origin but of 11q origin. In addition, were there no telomere deletion type? Also, were the interstitial telomere signals at the breakpoint region on 9p on the der(9)t(9;11) chromosome or where there only telomere signals at the end of this chromosome? If there were interstitial telomere signals on der(9)t(9;11), as can been seen in jumping translocations, did these signals have different telomere phenotypes and were they included in calculations? Were there any telomeric differences between short and long arms, especially regarding chr. 9 and 11?
It is indicated that mother and proband had no non-clonal aberrations and micronucleus formation compared with the farther (line 213, Figure 6); what were the observed frequencies of these abnormalities in the individuals and controls? Were these tests repeated? It could be that the father had stressed cells (e.g. due to viral infections, culture artifacts) compared with the mother and proband? A more general question, were the used white blood cells from each of the three individuals cultured, and if so, was it with or without PHA-mitogen before performing telomere analyses.
Does the breakpoint on the duplicated and translocated 11q harbor any gene? Are there repetitive sequences in this area which could influence the chance of the chromosomal event?
Any indications as to whether the chromosomal imbalance event in the offspring toke place in germinal cells (father or mother) or early/late post-fertilization? How well do sperm telomere length correlate with WBC telomere length? Such information might give clues to possible transmittance as studies have shown that offspring telomere length is significantly correlated with paternal, but not maternal telomere length (Njajou OT et al, PNAS, 104(29), 2007).
The authors conclude that their data strongly suggest that telomere dysfunction plays a central role in the formation of a particular unbalanced chromosome rearrangement via end-to-end fusion and breakage-fusion-bridge cycles. A more hypothetical voice is suggested since further studies will be needed to demonstrate such a hypothesis.
Minor issues:
Figure 3A: According to the insert, green circles are controls, blue triangle is father, red square is mother, and purple cross is proband which is not consistent with figure legend. Please clarify.
Line 32 and Figure 5: “t(9;11)(p24;q22.3)” should be “der(9)t(9;11)(p24;q22.3)”
Figure legend to Figure 6: “multiple chromosomal aberrations” should be “multiple non-clonal aberrations” to be consistent with the introducing body text.
Lines 114-115: After capture of telomere and centromere signals the M-FISH technique was applied. From this it is not clear whether the telomere and centromere signals were removed or not? Please clarify.
Did the couple have any previous pregnancies or spontaneous abortions?
Lines 334, 351, 366 and 370 in Reference section: The journal names are missing.
Author Response
Thank you very much for these comments
With respect to telomere analysis in section 3.2: What is the p-values for the proband and his parents to determine significance? In the figure legend to Fig. 3A it is stated that the telomere length of the father is within lower limits while in the body text it is stated to be significantly shorter. Please clarify.
p-values are added. The text and the legend to figure 3 are now concordant.
With respect to telomere phenotypes (Fig. 4): Telomere aberrations are divided into three types but in Fig. 4 only two (doublets and loss) of these are depicted – what about the deletion type?
It would be helpful to show representative images of the different subtypes from each of the analyzed individuals. At the cellular and chromosomal level what is the impact of the different aberrant telomere subtypes and what do they reflect?
Very low frequencies of telomere deletions were observed. For these reasons, we have deleted this aberration in figure 3. Additional images of metaphases from each individual have been added to figure 4
The impact of these aberrations in the chromosomal instability process has been described in the second paragraph of 3.2. telomere analysis (Results).
With respect to telomere analysis of individual chromosomes (Fig. 5): The x-axis indicates only normal chromosomes, which seems confusing because the proband has an unbalanced der(9)t(9;11), i.e. one normal chromosome 9 and one derivative. Further, it is not clear how this was handled when calculating frequency of telomere loss and doublet formation - the telomere on the der(9)t(9;11) p-arm is not of 9p origin but of 11q origin. In addition, were there no telomere deletion type? Also, were the interstitial telomere signals at the breakpoint region on 9p on the der(9)t(9;11) chromosome or where there only telomere signals at the end of this chromosome? If there were interstitial telomere signals on der(9)t(9;11), as can been seen in jumping translocations, did these signals have different telomere phenotypes and were they included in calculations? Were there any telomeric differences between short and long arms, especially regarding chr. 9 and 11?
The comment concerning the implication of der(9)t(9;11) on the frequencies of telomere loss and doublets in the proband is very interesting. For this reason and for the clarity of the paper the frequency of telomere loss and doublets detected in the lymphocytes of proband have been deleted in figure 5.
Concerning interstitial telomere signals on der(9)t(9;11), no telomeric signals were detected at the breakpoint region on 9p on the der(9)t(9;11).
The data concerning the implication of short and longs arms of chromosomes are not shown, but short arm of chromosomes 9 and long arm of chromosome11 exhibited a higher rate of telomere aberrations as shown in figure 6A.
It is indicated that mother and proband had no non-clonal aberrations and micronucleus formation compared with the farther (line 213, Figure 6); what were the observed frequencies of these abnormalities in the individuals and controls? Were these tests repeated? It could be that the father had stressed cells (e.g. due to viral infections, culture artifacts) compared with the mother and proband? A more general question, were the used white blood cells from each of the three individuals cultured, and if so, was it with or without PHA-mitogen before performing telomere analyses.
Non-clonal aberrations as well as micronucleus formation were induced by genotoxic stress. In comparison with healthy donors a significant difference was observed only in father. The three subjects were subjected to identical treatments, and culture conditions included PHA treatment. A large study in the implication of these aberrations and their utilities in post natal analysis is ongoing.
Does the breakpoint on the duplicated and translocated 11q harbor any gene? Are there repetitive sequences in this area which could influence the chance of the chromosomal event?
Yes, 11q22.3 contains a lot of repeated sequences in addition to the 107 genes.
Any indications as to whether the chromosomal imbalance event in the offspring toke place in germinal cells (father or mother) or early/late post-fertilization? How well do sperm telomere length correlate with WBC telomere length? Such information might give clues to possible transmittance as studies have shown that offspring telomere length is significantly correlated with paternal, but not maternal telomere length (Njajou OT et al, PNAS, 104(29), 2007).
Sperm telomere length was not investigated in this study, because there were no clinical indication to perform this test. We will need additional ethic permission to perform analysis of sperm telomere length. However, previous studies have demonstrated significant correlation between telomere length in sperm and in circulating lymphocytes
Concerning the transmission of telomere length and aberrations: By employing our technique on a large cohort of families with genetic diseases we have demonstrated that telomere length in the offspring can be related to paternal and maternal telomere length (manuscript in preparation). For this reason, we have not discussed this hypothesis.
The authors conclude that their data strongly suggest that telomere dysfunction plays a central role in the formation of a particular unbalanced chromosome rearrangement via end-to-end fusion and breakage-fusion-bridge cycles. A more hypothetical voice is suggested since further studies will be needed to demonstrate such a hypothesis.
I agree with reviewer, genomic analysis of this region will advance our knowledge and the implication of some mutations in telomere profiles as well as clinical syndromes. Such a genomic analysis is ongoing.
Minor issues:
Figure 3A: According to the insert, green circles are controls, blue triangle is father, red square is mother, and purple cross is proband which is not consistent with figure legend. Please clarify.
Thank you for this comment. The legend is corrected accordingly.
Line 32 and Figure 5: “t(9;11)(p24;q22.3)” should be “der(9)t(9;11)(p24;q22.3)”
Corrected now.
Figure legend to Figure 6: “multiple chromosomal aberrations” should be “multiple non-clonal aberrations” to be consistent with the introducing body text.
Modified now.
Lines 114-115: After capture of telomere and centromere signals the M-FISH technique was applied. From this it is not clear whether the telomere and centromere signals were removed or not? Please clarify.
The image of telomere and centromere staining followed by MFISH technique was shown in figure3 and 6. We removed telomere and centromere signal before MFISH staining.
Did the couple have any previous pregnancies or spontaneous abortions?
No spontaneous abortion has been declared and the couple has a children without any complication
Lines 334, 351, 366 and 370 in Reference section: The journal names are missing.
Corrected now
Reviewer 3 Report
The paper by Kacher and colleagues present a case study of a of a six-month-old boy with a chromosomal translocation, displaying telomere’s aberration in the chromosomes interested by the rearrangement. The paper is very well written and addresses a relevant issue concerning how telomere aberrances frequency impact predisposition to chromosomal diseases. They also raise a fundamental question regarding the importance of telomere aberrances diagnosis in the prenatal genetic screenings. Nevertheless, there are some points that need to be addressed before publication.
-For each figure (from 2 to 6) the authors need to specify which kind of cells have been used and how many metaphases/cells have been analyzed.
-Figure 4: it is not clear what is represented in the histograms, how frequency was calculated? Is it the number of aberrations per chromosome? How many metaphases have been analyzed per sample?
Figure 5: the control sample is not shown, here again it is not clear how many metaphases have been analyzed. It would be helpful to show a picture of pan telo/centromere analysis and M banding.
Author Response
Thank you very much for these comments
-For each figure (from 2 to 6) the authors need to specify which kind of cells have been used and how many metaphases/cells have been analyzed.
A specific paragraph as to the lymphocyte culture and metaphase preparation has been added (method section). The type of cells and the number of metaphase analyzed have been added in the figure legends.
-Figure 4: it is not clear what is represented in the histograms, how frequency was calculated? Is it the number of aberrations per chromosome? How many metaphases have been analyzed per sample?
Additional information have been added in figure 4 (5 now) concerning the number of metaphases and of the aberrations scored
Figure 5: the control sample is not shown, here again it is not clear how many metaphases have been analyzed. It would be helpful to show a picture of pan telo/centromere analysis and M banding.
Additional information has been added in figure 5 (6 now) concerning the number of metaphases and the aberrations scored. The control samples are not shown because the analysis of a large cohort of healthy donors is ongoing. The aim of this figure is to demonstrate the higher rate of telomere aberrations in the chromosomes implicated in the structural aberrations in the proband. For these reason we have deleted the results of the proband.